# A PPR Protein RFCD1 Affects Chloroplast Gene Expression and Chloroplast Development in *Arabidopsis*

**DOI:** 10.3390/plants14060921

**Published:** 2025-03-15

**Authors:** Tianming Tan, Shengnan Xu, Jiyun Liu, Min Ouyang, Jing Zhang

**Affiliations:** 1National Key Laboratory of Crop Genetic Improvement, Huazhong Agricultural University, Wuhan 430070, China; 17720303520@163.com (T.T.); shengnanxu1127@163.com (S.X.); 17320504651@163.com (J.L.); ouyangmin@mail.hzau.edu.cn (M.O.); 2Hubei Hongshan Laboratory, Huazhong Agricultural University, Wuhan 430070, China; 3College of Life Science and Technology, Huazhong Agricultural University, Wuhan 430070, China; 4College of Life Sciences, Hubei University, Wuhan 430062, China

**Keywords:** *Arabidopsis*, chloroplast development, Regulation Factor of Chloroplast Development 1 (RFCD1), pentatricopeptide repeat (PPR) protein, gene expression

## Abstract

Chloroplast development is a highly complex process, involving many regulatory mechanisms that remain poorly understood. This study reports a novel PPR protein, RFCD1 (Regulation Factor of Chloroplast Development 1). Fluorescence localization analysis reveals that the N-terminal 60 amino acids of RFCD1 fused with GFP protein specifically direct the protein to the chloroplast. The knockout mutant of RFCD1 is embryo-lethal. *RFCD1* RNA interference (RNAi) transgenic lines display chlorosis phenotypes and abnormal chloroplast development. Quantitative real-time PCR (qRT-PCR) showed that the expression levels of the plastid-encoded RNA polymerase (PEP) genes were significantly decreased in the RNAi lines. Furthermore, RNA blotting results and RNA-seq data showed that the processing of plastid rRNA was also affected in the RNAi lines. Taken together, these results indicate that RFCD1 might be involved in chloroplast gene expression and rRNA processing, which is essential for chloroplast development in *Arabidopsis*.

## 1. Introduction

The process of photosynthesis not only converts carbon dioxide into energy-rich organic compounds, but also produces oxygen, thus fundamentally sustaining the global ecosystem [1]. Photosynthesis takes place in chloroplasts, providing the necessary energy and metabolic substances for plant growth and development. Proper chloroplast development is a prerequisite for the plant’s ability to conduct photosynthesis.

The biogenesis and developmental processes of chloroplasts are highly complex and ordered. They are controlled not only by the transcription and translation of nuclear and chloroplast genes but also by the retrograde signaling between the chloroplast and nucleus [2]. The transcription of chloroplast genes relies on the coordinated action of plastid-encoded RNA polymerase (PEP) and nuclear-encoded RNA polymerase (NEP) [3]. PEP is a bacterial-type polymerase composed of multiple subunits and is responsible for about 80% of chloroplast gene transcription [4]. PEP consists of four core subunits which are encoded by the chloroplast genes *rpoA*, *rpoB*, *rpoC1*, and *rpoC2*, as well as a sigma-like transcription factor encoded by the nuclear genome that recognizes promoters [5,6]. During the early stage of chloroplast development, the nuclear-encoded NEP plays a major role by preferentially transcribing the core subunits of PEP, thereby establishing the chloroplast transcription and translation system [7]. Recent studies have revealed the cryo-EM structure of the chloroplast transcriptional machinery, the PEP-PAP supercomplex in *Spinacia oleracea*, *Sinapis alba*, and *Nicotiana tabacum* [8,9,10,11]. The PEP core has a “crab claw” appearance with two arms: a lower arm (the lobe-protrusion-claw) and an upper arm (the clamp-claw). Besides the PEP core, more than a dozen nuclear-encoded eukaryotic-origin PEP-associated proteins (PAPs) tightly interact with the PEP core. The 13 PAPs can be grouped into five clusters according to their location and potential functions. The first cluster includes PAP1, PAP7, and PAP11, which are located on the upper arm. This cluster may integrate multiple enzymatic activities and could additionally serve as a platform for the binding of additional factors that mediate co-transcriptional processes. The second cluster contains PAP5 and PAP8, which are positioned at the intersection of the two arms. This cluster fulfills structural roles in the PEP complex by facilitating interactions between the core polymerase subunits and more peripheral PAPs. The third cluster contains PAP3 and PAP14, which are located on the lower arm and form a scaffold for the lobe-protrusion-claw, providing the docking site for the PAP4–PAP9 heterodimer. This cluster can stabilize the PEP core. These three clusters are also called the scaffold module, which shield about half of the PEP surface, and interlink multiple domains of the PEP core. The fourth and fifth clusters consist of PAP6–PAP10s–PAP13 and PAP4–PAP9, respectively. The fourth cluster is also called the regulation module. This module may confer redox-activity that makes tight interactions with subunits of both the PEP core and the scaffold module, and likely plays a role in assembling the complex and maintaining its stability. The fifth cluster is also called the protection module. This module may confer superoxide-detoxification activity that can protect the PEP complex from oxidation damage. Besides these 13 PAPs, PAP2 is called the RNA module that binds RNA in a sequence-specific manner, while PAP15 has the potential to contribute to the DNA-binding activity of PEP. These clusters facilitate the assembly of the transcription apparatus, protect the complex from oxidative damage, and likely couple gene transcription with RNA processing, ensuring the precise execution of plastid gene expression [8,9,10,11]. Genetic disruption of individual *PAP* genes leads to albino or pale green phenotypes, severely impaired chloroplast development, and reduced PEP-dependent transcription, suggesting that PEP transcription activity is essentially dependent on each PAP [8,12]. Among them, RCB (Regulator of Chloroplast Biogenesis) [13] and NCP (Nuclear Control of PEP activity) [14] are all dual-localized proteins in the nucleus and in chloroplasts and play pivotal roles in mediating nuclear-plastid communication. RCB and NCP are required for the formation of large photobodies without affecting phyB protein level or nuclear localization, and are also required for PEP assembly. However, both of them are not components of the photobody in the nucleus or the PEP complex in the chloroplasts. RCB/NCP-dependent photobody formation and PIF degradation in the nucleus triggers the assembly and activation of the PEP complex during chloroplast biogenesis [13,14,15].

The translation mechanisms of chloroplasts in plants have undergone high specialization during evolution, differing from those of bacteria or other eukaryotic organelles. Protein translation in chloroplasts occurs on 70S ribosomes which consist of two subunits: the 30S and 50S ribosomal subunits. The 30S subunit is composed of 16S rRNA and approximately 20 ribosomal proteins, while the 50S subunit contains 23S rRNA, 5S rRNA, 4.5S rRNA, and about 30 ribosomal proteins [16]. The biogenesis of ribosomes is a highly complex process that involves the proper processing, covalent modification, and restructuring of various rRNA species, as well as the coordinated assembly of ribosomal proteins in a spatiotemporal manner [17,18]. Numerous chloroplast ribosome assembly factors involved in rRNA processing, maturation, restructuring, and coordinated insertion of ribosomal proteins into the functional ribosomal complex have been discovered and characterized in recent years [19]. Genome-wide analyses have unraveled the suborganellar localization of translation and its participation in controlling the developmental program of chloroplast gene expression [20]. These proteins include nuclear-encoded pentatricopeptide repeat (PPR) motif proteins and octotricopeptide repeat (OPR) proteins, ribonucleases and RNA-helicases in rRNA maturation, GTPases in ribosome assembly, additional proteins aiding in rRNA processing, chloroplast RNA chaperones, and assembly factors interacting with ribosomal subunits [19].

PPR proteins constitute one of the largest protein families in land plants, with more than 400 members in most species, and are well known for their roles in chloroplast and mitochondrial development [21,22]. A typical PPR protein is targeted to mitochondria or chloroplasts, binds one or several organellar transcripts, and influences their expression by altering RNA sequence, turnover, processing, or translation [22]. PPR is a domain characterized by a repetitive structure of triangular pentapeptides arranged in tandem. The modular nature of helical repeat proteins may have provided the basis for the rapid evolution of new sequence specificities when the families expanded and were recruited to assist RNA processing in plant organelles [23]. Based on the number of amino acids in each motif, PPR proteins can be classified into two types: P-type and PLS-type [24]. In higher plants, there are a large number of genes encoding PPR proteins. The *Arabidopsis* genome contains approximately 450 PPR genes, while the rice (*Oryza sativa*) genome includes about 477 PPR genes [25]. Increasing evidence suggests that members of the PPR family play an essential regulatory role in plant growth and development [26,27,28]. Members of the PPR protein family in vascular plants are required to stabilize certain mRNAs, to facilitate splicing of specific introns, or to select particular C targets for the editing apparatus [23]. Loss of various PPR proteins can produce a similar phenotype with respect to chloroplasts, such as pale-yellowish pigmentation, altered PSII biogenesis and formation of grana thylakoids, impaired chlorophyll synthesis, and severe defects in photosynthesis [29]. Additionally, the PPR protein family plays a crucial role in the post-transcriptional regulation of chloroplast RNA. These proteins primarily function by specifically binding to chloroplast RNA, mediating RNA editing, splicing, stability regulation, and translation, thereby influencing the precision and efficiency of chloroplast gene expression [6]. The PPR proteins involved in editing belong to a subclass. Two functional domains—the extension (E) domain and the DYW domain—have been identified in this subclass of PPR proteins [30]. Among them, YS1 (YELLOW SEEDLING 1) has a C-terminal DYW motif that edits *rpoB* transcripts at site 25992, allowing fully active RpoB PEP subunits to be translated [31].

The role of PPR proteins is not limited to the processing of individual RNA molecules; rather, they function as key regulators within the entire chloroplast RNA metabolism network. They are extensively involved in mRNA splicing, maturation, translational activation, as well as rRNA and tRNA processing, ensuring the accuracy and efficiency of chloroplast protein synthesis [32]. Consequently, the loss of function of these factors often results in severe chloroplast developmental defects, such as impaired photosynthetic capacity, reduced chlorophyll synthesis, abnormal chloroplast structure, and even seedling lethality [33]. These observations highlight the central role of the PPR protein family in maintaining chloroplast homeostasis, demonstrating that their precise regulation is essential for chloroplast development and the overall physiological functions of plants. However, to date, the biological function of many PPR proteins in chloroplast development remains unknown.

In this study, we have identified a new PPR protein, RFCD1 (Regulation Factor of Chloroplast Development 1), that participates in chloroplast development. The homozygous knock-out mutant has an embryo-lethal phenotype, while RNAi mutant plants display cotyledon chlorosis and yellowing of true leaves. The mutation of RFCD1 also leads to impaired chloroplast development, deficient accumulation of photosynthetic proteins, decreased expression levels of PEP-dependent genes, and abnormal rRNA processing. These results suggest that RFCD1 plays a crucial role in chloroplast development during seedling growth.

## 2. Results

### 2.1. RFCD1 Is Essential for Plant Growth

Since loss of PPR proteins often leads to an embryo-lethal phenotype, knockout mutants cannot be used for determining their roles in chloroplast development. We therefore first tested all unknown chloroplast PPR proteins via the CRISPR-Cas9 system to test the embryo-lethal phenotypes in the resulting knockout mutants. Then, we constructed the corresponding knockdown mutants via RNA interference, thereby circumventing the embryonic lethality caused by the complete knockout, and finally selected the RNAi mutants with defects in early chloroplast development for further study. Among these mutants, we found that no homozygous knockout of the *AT2G01860* gene was obtained, indicating that the loss of this gene may lead to an embryo-lethal phenotype. We then dissected mature siliques of *AT2G01860* heterozygotes. Some ovules in the mature siliques of *AT2G01860* heterozygotes appeared white, and the ratio of normal to white embryos was 3:1; while in wild-type siliques, all embryos developed normally (Figure 1A). Thus, we renamed the *AT2G01860* gene as Regulation Factor of Chloroplast Development 1 (*RFCD1*).

To further investigate the function of RFCD1, we constructed *RFCD1* RNA interference plants and obtained approximately 90 transgenic lines. Among them, 48 showed chlorosis phenotypes. Then, we selected three RNAi transgenic lines with different chlorosis phenotypes in cotyledons and true leaves, named *RFCD1*-RNAi 1–3 (Figure 1B). Reverse transcription PCR (RT-PCR) was then performed to measure the expression levels of RFCD1 in wild-type and these three RNAi seedlings. The results showed that the expression levels of RFCD1 were consistent with the phenotypes of RNAi plants (Figure 1B–D). *RFCD1*-RNAi-1 was chosen for subsequent analysis, as this line had the most severe phenotype.

### 2.2. RFCD1 Encodes a PLS-Type Chloroplast PPR Protein

The *RFCD1* gene encodes a protein of 486 amino acids, with an estimated molecular weight of 55.6 kDa. The first 60 amino acids at the N-terminus of RFCD1 are predicted chloroplast transit peptide (predicted by TargetP, http://ppdb.tc.cornell.edu/) (accessed on 12 March 2025) (Figure 2A). To investigate the subcellular localization of RFCD1, the predicted chloroplast transit peptide of RFCD1 was fused with a green fluorescent protein (GFP), and the constructed vector was transiently expressed in *Arabidopsis* mesophyll protoplasts. The green fluorescence of the fusion protein could be merged with the chlorophyll autofluorescence, suggesting that RFCD1 is localized to the chloroplasts (Figure 2B). To further confirm the localization of RFCD1, we transformed the full-length RFCD1 protein fused with a GFP tag into *Arabidopsis* mesophyll protoplasts. The cytosol, nucleus, and chloroplasts were isolated from these protoplasts, and then subjected to immunoblot analysis. The results showed that the GFP-tagged RFCD1 protein was only detected in the chloroplast fraction (Figure 2C), confirming the chloroplast localization of RFCD1.

Sequence analysis revealed that the RFCD1 protein contains five tandem PPR repeats in the C-terminal part (Figure 2A and Appendix A), and it belongs to the PPR PLS subgroup of the PPR protein family [34]. To examine the conservation of RFCD1 across different species, we searched RFCD1 homologs in dicots, monocots, gymnosperms, and mosses. Sequence alignment showed that RFCD1 exhibits a high level of similarity in *Populus trichocarpa* (*Potri.005G252300.1*, 62% identity) and *Gossypium hirsutum* (*V6Z11_D08G301300*, 60% identity), while it showed a relatively low level of similarity in *Physcomitrella patens* (*Mp_5g22200*, 36% identity) and *Ceratopteris richardii* (KP509_10G067400, 36% identity) (Appendix A). According to the sequence alignment, we created a phylogenetic tree of these homologs to reveal the relationship among these species (Figure 2D). The above results indicate that RFCD1 is conserved in most plants but with a higher sequence identity in Spermatophyta.

### 2.3. Deficiency of RFCD1 Causes Abnormal Chloroplast Development

To investigate the role of RFCD1 in chloroplast development, we analyzed the ultrastructure of chloroplasts in cotyledons and true leaves from 10-day-old seedlings (Figure 3). In wild-type plants, chloroplasts were oval-shaped, and the thylakoid structures were clearly discernible and well developed. In contrast, chloroplasts in the *RFCD1*-RNAi-1 mutants were smaller than those of the wild type, and contained rudimentary thylakoids. These results indicate that the downregulation of *RFCD1* leads to abnormal chloroplast development, which affects the size of chloroplasts, the content and organization of thylakoids.

### 2.4. Knockdown of RFCD1 Affects the Accumulation of Photosynthetic Proteins

Given the reduced chlorophyll content and defective chloroplast structure of the *RFCD1*-RNAi mutants, we hypothesized that the photosynthetic complexes might also be affected. Accordingly, proteins from cotyledons and whole seedlings of 7-day-old wild-type and *RFCD1*-RNAi-1 seedlings were extracted, and equal amounts of protein were subjected to immunoblot analysis to detect the levels of subunits of various photosynthetic complexes, including Photosystem II (D1, LHCII), Photosystem I (PsaA/B, PsaN, LhcaI), and Cytochrome *b*_6_*f* complex (Cyt-f). The results showed that the accumulation of these photosynthetic proteins was significantly reduced in both cotyledons (Figure 4A,B) and whole seedlings (Figure 4C,D) of *RFCD1*-RNAi-1. Moreover, the levels of photosynthetic proteins in the cotyledons were significantly decreased compared to the whole seedlings. These results suggested that RFCD1 plays a crucial role in the accumulation of photosynthetic proteins in plants.

### 2.5. RFCD1 Mutation Leads to an Altered Plastid Gene Expression

Plastid gene expression plays a crucial role in plastid development. We used qRT-PCR to measure the abundance of different chloroplast transcripts in cotyledons and true leaves of 7-day-old WT and *RFCD1*-RNAi-1 seedlings (Figure 5A,B). The results showed that, compared to the wild type, the transcription levels of all of the class I genes (primarily transcribed by PEP) were downregulated in both cotyledons and true leaves of *RFCD1*-RNAi-1 mutants. On the other hand, the expression levels of all of the class III genes, which are mainly transcribed by NEP, were upregulated in *RFCD1*-RNAi-1 mutants. Additionally, the expression levels of class II genes, which are co-transcribed by both PEP and NEP [35], displayed differential regulation patterns. We also examined some chloroplast genes that have not been fully classified (Appendix A). The transcript levels of some of these genes were upregulated, while others were downregulated. These results suggested that the developmental defects observed in *RFCD1*-RNAi mutants may be due to the impairment of PEP activity.

### 2.6. Transcriptome Analysis of RFCD1-RNAi-1 Seedlings

To further investigate the effect of RFCD1 deficient in gene expression, we performed RNA-Seq analysis to compare the transcriptomes of *RFCD1*-RNAi-1 and wild-type plants. After removing invalid reads, we obtained 44836582, 44662430, and 51293706 clean reads from *RFCD1*-RNAi-1 seedlings, and 47215108, 45429826, and 40927384 clean reads from WT plants. The percentages of Q20 sequences (nucleotides with quality values larger than 20 in reads) for *RFCD1*-RNAi-1 and WT were 98.03% and 98.09%, respectively, and the GC (guanine and cytosine) contents for *RFCD1*-RNAi-1 and WT were 45.42% and 45.74%, respectively (Appendix A).

Volcano plots showed the differential gene expressions (DEGs) between *RFCD1*-RNAi-1 and WT plants in three biological replicates. A total of 3680 DEGs were identified, with 2047 upregulated and 1633 downregulated genes (Appendix A). Functional categorization of DEGs based on the Gene Ontology (GO) database revealed that the most abundant category was biological processes (Appendix A).

Based on the GO analysis results, a total of 455 significantly enriched terms were identified, with 287 terms being significantly enriched in biological processes (BPs). The main enriched terms in the top 50 include processes related to photosynthesis (marked by ‘▲’), particularly light capture, suggesting that RFCD1 may play a role in photosynthesis (Figure 6A). Additionally, the significant enrichment of processes in chloroplast RNA metabolism (marked by ‘△’), such as chloroplast RNA processing, indicated that RFCD1 may function in RNA processing and modification (Figure 6A). Other significantly enriched terms also related to amino acid metabolism (marked by ‘★’), indirectly suggest that RFCD1 may affect the amino acid metabolism in the chloroplast (Figure 6A). In terms of cellular components (CC), 94 terms were enriched, with the significantly enriched term, such as chloroplast nucleoid and ribosome, thylakoid, stroma, and envelope (marked by ‘△’ and ‘▲’), further implying the impact of RFCD1 in chloroplast gene expression and biogenesis (Figure 6B). At the molecular function (MF) level, 74 terms were significantly enriched, covering functions such as chlorophyll binding (marked by ‘▲’), rRNA and RNA binding (the terms are marked by ‘△’), raising the possibility that RFCD1 may participate in ribosome assembly (Figure 7). Furthermore, the enrichment of tetrapyrrole and vitamin B6 binding terms (marked by ‘★’) further confirms the potential function of RFCD1 in chlorophyll biosynthesis and amino acid metabolism in the chloroplast (Figure 7).

In the Kyoto Encyclopedia of Genes and Genomes (KEGG) pathway analysis, the top enriched pathways include photosynthesis–antenna proteins, amino acid metabolism (valine, leucine, and isoleucine degradation; glycine, serine, threonine, cysteine, and methionine metabolism), and ribosome with a large number of genes enriched, consistent with the GO analysis results (Figure 8A). Taken together, these results suggest that RFCD1 may play a role in the expression of RNA processing and ribosomal genes, and genes related to the fundamental metabolic processes or mainly occurring in chloroplasts, such as photosynthesis and amino acid metabolism.

We further analyzed the expression levels of genes related to photosynthesis and found that the expression levels of most antenna proteins were significantly reduced (Figure 8B). Besides the antenna proteins, there were in addition 17 other genes related to photosynthesis. Photosynthesis is a complicated process that requires high coordination of gene expression between nuclear and chloroplast genomes. Among these DEGs, we found that the expression levels of chloroplast-encoded photosynthesis proteins were downregulated, such as *psbA*, *psbB*, *atpB*, and *atpF*, while most of the nuclear-encoded photosynthesis proteins which are involved in the four major complexes of the light reactions were upregulated, such as genes encoding PsbP-like protein 1 (PPL1), PsbQ-like 1 and 2 (PnsL3 and PnsL2), NPQ4, PSB28, FD1, and FDC2. Additionally, amongst the photosynthetic proteins participating in the dark reactions, the expression levels of genes encoding ribulose bisphosphate carboxylase (small subunit) family proteins (RBCS-1B, RBCS-2B, and RBCS-3B) were upregulated which are also encoded by the nuclear genome (Appendix A).

In addition to photosynthesis, processes related to chloroplast gene expression were also affected. The expression levels of nuclear-encoded chloroplast-localized proteins of the ribosomal large subunit were generally increased, whereas those of the chloroplast-encoded chloroplast-localized proteins of the ribosomal large subunit were significantly reduced. By contrast, the expression levels of the proteins of the ribosomal small subunit showed a divergent pattern, with some of them increasing and others decreasing (Figure 8C). These RNA-seq results, combined with partial validation by qRT-PCR, indicate that RFCD1 is critical for photosynthesis and chloroplast development, and participates in chloroplast gene expression, such as the establishment of the PEP activity, chloroplast RNA processing, and plastid ribosome accumulation.

### 2.7. Accumulation of Chloroplast rRNAs Is Impaired in RNAi Lines

Chloroplast ribosomal RNAs are co-transcribed as a single RNA precursor that contains 16S rRNA, two tRNAs, 23S, 4.5S, and 5S rRNAs (Figure 9A). After transcription, the precursor transcript undergoes a series of maturation processing events [36]. During this process, a 23S–4.5S bi-cistronic RNA (3.2 kb) undergoes endonucleolytic cleavage to produce a mature 4.5S rRNA and a 23S precursor (2.9 kb). This 2.9 kb precursor finally produces three species of RNA (1.3 kb, 1.1 kb, and 0.5 kb), while the mature 16S rRNA (1.5 kb) is processed from the 16S precursor RNA (1.7 kb) [37].

To investigate whether RFCD1 affects processing of chloroplast rRNA, total RNA from the leaves of wild-type and *RFCD1*-RNAi-1 seedlings was extracted to examine the role of RFCD1 in this process. We first performed denaturing agarose gel electrophoresis followed by ethidium bromide staining, and found that in the *RFCD1*-RNAi-1 mutants, the levels of chloroplast 16S rRNA (1.6 kb) and 23S rRNA degradation products (1.1 kb) were lower compared to the wild type. (Figure 9B).

To further explore the function of RFCD1 in chloroplast rRNA processing, RNA blot analysis of the different rRNA transcripts was performed using rRNA-specific probes in Figure 9B. Higher transcript levels of the 1.7-kb 16S rRNA precursor, 3.2-kb, 2.9-kb, 2.4-kb, and 1.8-kb 23S precursor, were detected in the *RFCD1*-RNAi-1 mutants, whereas the levels of the 1.5-kb 16S mature rRNA and 1.3-kb 23S mature rRNA were largely decreased (Figure 9C). We also detected the *psbA* transcript level using a *psbA*-specific probe and found that the transcript level of *psbA* was decreased, consistent with the qRT-PCR and RNA-seq results (Figure 9C). Taken together, these results suggested that RFCD1 plays a crucial role in the processing of plastid rRNAs.

## 3. Discussion

Chloroplasts are the primary sites of photosynthesis in plants, and their proper development is essential for plant survival. The formation of functional chloroplasts relies on gene expression in both the nucleus and plastid. Chloroplast gene expression is regulated by numerous factors, particularly by nuclear-encoded chloroplast-localized proteins [38,39]. It has been shown that many embryo-lethal mutants often exhibit defects in chloroplast development and function [40]. Increasing evidence suggests that many PPR proteins are crucial for plastid gene expression and development. PPR protein Early Chloroplast Development 1 (ECD1) functions as an RNA-editing factor of *rps14*-149 in plastids and is required for early chloroplast development in seedlings [41]. *Arabidopsis* Early Chloroplast Development 2 (ECD2) and rice Young Leaf White Stripe (YLWS) are responsible for group II intron splicing and chloroplast gene expression during early leaf development [42,43].

In this study, we identified a new PPR protein in *Arabidopsis*, RFCD1, which plays a crucial role in chloroplast biogenesis and plant growth. Subcellular localization analysis showed that RFCD1 is located in chloroplasts. Further observation of chloroplast ultrastructure revealed that the size of chloroplasts in *RFCD1*-deficient mutants was smaller than in wild type and thylakoids in *RFCD1*-deficient mutants were not fully developed, suggesting that the white or chlorosis phenotype in these mutants is likely due to defective chloroplast development. Moreover, the complete loss of RFCD1 leads to embryo lethality, and the RNAi lines display phenotypes such as cotyledon bleaching, true leaf yellowing, and a significant reduction in chlorophyll content and the accumulation of several photosynthetic proteins, indicating the critical role of RFCD1 in chloroplast biogenesis and plant survival.

Considering that plastid gene expression and the expression of nuclear-encoded chloroplast proteins are tightly linked to the developmental status of chloroplasts [44], we observed a significant decrease in the transcript levels of class I genes (PEP-dependent) and a significant increase in the transcript levels of class III genes (NEP-dependent) by qRT-PCR analysis. This suggests that the early arrest of chloroplast development in *RFCD1*-deficient mutants is likely due to a reduction in PEP activity. Similarly, a defect in the function of other PPR proteins Pigment-Defective Mutant 2 (PDM2), Pigment-Defective Mutant 3 (PDM3), or Seedling Lethal 1 (SEL1) also leads to an impairment of PEP activity [45,46,47].

In addition to transcriptional regulation, translational regulation is an important step of gene expression in chloroplasts. The chloroplast proteins encoded by plastid genes are synthesized by the 70S ribosomes within the chloroplasts. Proper accumulation of plastid ribosomal proteins and rRNAs is a prerequisite for assembling functional ribosomes and is essential for plastid development. A lack of some nuclear-encoded chloroplast proteins impairs the assembly and accumulation of chloroplast ribosomes. For example, PPR287 is a protein containing 10 PPR domains that does not participate in intron splicing in the chloroplast, but affects the stability of chloroplast rRNA [48]. Our RNA-seq analysis revealed that partial defects in RFCD1 lead to a reduction in the expression of antenna proteins and chloroplast-encoded components of the photosynthetic electron transport chain, as well as a significant decrease in chloroplast-encoded chloroplast-localized ribosomal large subunits. By contrast, the expression of nuclear-encoded chloroplast-localized ribosomal large subunits was generally upregulated. Chloroplast ribosomes mainly consist of the 50S large and the 30S small subunits. The 30S subunit is primarily composed of 16S rRNA and ribosomal proteins, while the 50S subunit consists of three types of rRNAs: 23S, 4.5S, 5S rRNA, and ribosomal proteins [45]. Our RNA blot analysis revealed a high level of accumulation of rRNA precursors and a significant decrease in mature 16S rRNA (1.5 kb) and 23S rRNA (1.3 kb) in the *RFCD1*-RNAi-1 line. Combined with the RNA-seq results above, this indicates that decreased accumulation of RFCD1 impairs chloroplast rRNA processing and ribosome assembly, which further affects chloroplast gene expression, chloroplast biogenesis, and photosynthesis.

Similarly, the DEAD-Box RNA Helicase AtRH7/PRH75, which is also involved in rRNA processing, accumulates higher levels of rRNA precursor under normal conditions compared to the wild type [49]. In contrast, the loss of PPR287 resulted in a pale-green phenotype, and RNA blotting results showed that the accumulation of both precursor and mature products of 23S rRNA was decreased in the *ppr287* mutants. Additionally, knockout of the chloroplast PPR proteins CDB1 (Chloroplast Development and Biogenesis 1) or CRP1 (Chloroplast RNA Processing 1) displayed a white phenotype, and RNA blotting analysis revealed that the amounts of 2.9-kb precursor and 1.1-kb mature product of 23S rRNA were all drastically reduced. Moreover, knockout of chloroplast OPR protein RAP resulted in chlorotic leaves with decreased 16S ribosomal RNA while 23S ribosomal RNA remained unchanged. These RNA-blot results differ from those of the *RFCD1*-RNAi-1 mutant, suggesting that although defects in these proteins, such as PPR287, CDB1, CRP1, or RAP, perturb chloroplast biogenesis, these proteins affect plastid rRNA processing and accumulation in distinct ways compared to that of RFCD1. Based on this, we infer that the impairment of gene expression and function of chloroplasts in the *RFCD1*-RNAi mutants may be due to perturbations in rRNA processing and ribosome assembly, as well as in PEP activity. This also explains why the accumulation of photosynthetic proteins is decreased in the *RFCD1*-RNAi lines. Moreover, RFCD1 also affects amino acid metabolism, which may be due to impaired chloroplast functions, though this requires further investigation.

Collectively, our study indicates that the *Arabidopsis* PPR protein RFCD1 is essential for chloroplast development. Complete loss of RFCD1 leads to embryo lethality. Deficiency of RFCD1 causes a white or chlorosis phenotype in cotyledon and true leaves and a significant reduction in the accumulation of photosynthetic proteins, and affects chloroplast ultrastructure. Moreover, the levels of PEP-dependent gene transcripts were significantly affected in *RFCD1*-RNAi mutants, while plastid rRNA processing was also defective which in turn may affect plastid ribosome biogenesis and lead to upregulated expression of most of ribosomal proteins. These processes interact and restrict one another, which likely contributes to the loss of chloroplast development in both cotyledons and true leaves. However, the precise or direct cause of this phenomenon remains unclear. Therefore, further research on the function of RFCD1 is necessary to fully understand the mechanisms underlying chloroplast development.

## 4. Materials and Methods

### 4.1. Plant Materials and Growth Conditions

The *Arabidopsis thaliana* wild-type (Columbia ecotype) seed was used in this experiment. Wild-type and mutant seeds were stratified at 4 °C for 2 days, surface-sterilized, and sown on MS medium supplemented with 2% sucrose. The plants were grown under 12 h light/12 h dark cycles with a photon flux density of 120 μmol m^−2^ s^−1^ at 22 °C.

### 4.2. Generation of RNAi Lines

For the RNAi lines, a fragment of *RFCD1* (330 bp) was subcloned into the cloning sites of pFGC5941 to create a binary vector. The recombinant plasmid was then transformed into the *Agrobacterium tumefaciens* strain GV3101, as described previously [50]. In addition, the floral-dip method was used to produce transgenic plants according to Clough and Bent (1998) [51]. The transgenic plants were selected on MS medium containing 50 μg/mL Basta. Transgenic Basta-resistant seedlings were sown as described before. Three transgenic lines named *RFCD1*-RNAi-1, *RFCD1*-RNAi-2, and *RFCD1*-RNAi-3 are presented in this study.

### 4.3. Chlorophyll Content Measurement

Chlorophyll content was measured to determine the chlorophyll levels in the cotyledons of 7-day-old *Arabidopsis* seedlings. An appropriate amount of fresh leaves (about 0.1 g) was taken, cut into small pieces, and placed in a test tube. Chlorophyll was extracted with 80% acetone and quantified using a UV2800 spectrophotometer (Unico, Dayton, NJ, USA) to measure the absorbance at OD646 and OD663. Chlorophyll content was determined according to the following formulas:Chl a (mg/L) = 12.21 × OD663 − 2.81 × OD646Chl b (mg/L) = 20.13 × OD646 − 5.03 × OD663Total Chl (mg/L) = 17.32 × OD646 + 7.18 × OD663

The final chlorophyll content was calculated based on the fresh weight (FW) of the sample using the following formula:Chl (mg/g FW) = C × V/W

C: Chlorophyll concentration (mg/L)

V: Volume of extraction solution (L)

W: Fresh weight of the sample (g)

Three biological replicates were analyzed for each sample, with three technical replicates per biological replicate [52].

### 4.4. Subcellular Localization of RFCD1-GFP Fusion and Visualization

A 60-amino acid N-terminal fragment of the RFCD1 protein was amplified and fused to GFP (green fluorescent protein). The GFP sequence was inserted into the pUC18-35S-sGFP vector as a reporter gene. The fusion vector was introduced into *Arabidopsis* mesophyll protoplasts via the polyethylene glycol (PEG)-mediated method [53]. After 14–18 h incubation of the transformed protoplasts, the fluorophore-labeled protein was observed using a confocal laser scanning system (LSM980; Carl Zeiss, Oberkochen, Germany). GFP fluorescence was excited at 488 nm using a confocal microscope, chloroplast autofluorescence was excited at 594 nm, and bright-field images were simultaneously captured, with all images processed and exported using Zeiss 2.3 software.

### 4.5. Isolation of Cytosol, Nucleus, and Chloroplast

For the isolation of cytosol and nucleus, the transformed protoplasts that overexpressed the full-length RFCD1 protein fused with GFP were collected and ground to a fine powder in liquid nitrogen and mixed with 2 volumes of lysis buffer (20 mM Tris-HCl, pH 7.4, 25% glycerol, 20 mM KCl, 2 mM EDTA, 2.5 mM MgCl_2_, 250 mM sucrose, and 1 mM PMSF). The homogenate was filtered through a 95- and 37-μm nylon netting successively. The flow-through was spun at 1500× *g* for 10 min, and the supernatant consisting of the cytosolic fraction was collected and mixed with SDS sample buffer (300 mM Tris-HCl, pH 6.8, 5% SDS, 0.5% bromophenol blue, 50% glycerol) and heated at 95 °C for 10 min. The pellet was washed four times with 5 mL of nuclear resuspension buffer NRBT consisting of 20 mM Tris-HCl, pH 7.4, 25% glycerol, 2.5 mM MgCl_2_, and 0.2% Triton X-100 [54]. The final pellet was mixed with 50 μL of SDS sample buffer and heated at 95 °C for 10 min.

Chloroplast isolation was performed as previously described with the following modifications. The transformed RFCD1-GFP protoplasts were harvested and resuspended in extraction buffer (1× GM: 1 mM Na-pyrophosphate, 50 mM HEPES, 0.33 M Sorbitol, 2 mM Na-EDTA, 1 mM MgCl_2_, 1 mM MnCl_2_). Centrifugation was at 4000× *g*, 4 °C for 5 min, then the supernatant was discarded. The pellet was carefully transferred onto a Percoll gradient (95% Percoll, 3% PEG4000, 1% BSA, 1% Ficoll 400K) using a wide-bore pipette tip. Centrifugation was at 4000× *g*, 4 °C for 30 min. The intact chloroplasts were collected from the 50–80% Percoll gradient interface, washed by adding 5 volumes of resuspension buffer (0.33 M Sorbitol, 50 mM HEPES-NaOH, pH 7.9), and the tube was gently inverted to remove the Percoll. Centrifugation was at 4000× *g*, 4 °C for 5 min, and the pellet containing intact chloroplasts was retained. The final pellet was mixed with 50 μL of SDS sample buffer and heated at 95 °C for 10 min. Five microliters of each fraction were loaded on an 10% SDS-PAGE gel for protein separation and immuno-blots using specific antibodies, including anti-actin antibody (ABclonal, Woburn, MA, USA, AC004), anti-H3 antibody (ABclonal, A2348), laboratory-produced anti-LHCII antibody, and anti-GFP antibody.

### 4.6. RNA Gel Blotting, RT-PCR, and Quantitative RT-PCR

RNA was isolated from the cotyledons of 7-day-old and true leaves of 10-day-old wild-type and *RFCD1*-RNAi seedlings using the Trizol RNA extraction reagent (KR009-1, Kukkin Biotech, Kunming, China). RNA was separated on 1.2% (*w*/*v*) agarose-formaldehyde gels, transferred to a nylon membrane (RPN303B, Cytiva, Marlborough, MA, USA), and hybridized with a probe labeled with dUTP according to the manual from Roche Applied Science (https://elabdoc-prod.roche.com) (accessed on 12 March 2025). The probe was prepared by PCR amplification and labeled with Anti-Digoxigenin-11-dUTP (Roche, Basel, Switzerland, 11093088910). The signals from the secondary conjugated antibodies were detected using the Ready-use CDP-Star solution (LA10864, Meilunbio, Dalian, China).

RNA concentration was measured using a NanoDrop2000 spectrophotometer. cDNA was synthesized from total RNA using the TransScript One-Step gDNA Removal and cDNA Synthesis SuperMix Kit (6210A, Takara, Kusatsu, Japan). cDNA samples served as templates for qRT-PCR analysis. For qRT-PCR, ArtiCanATM SYBR qPCR Mix (Low ROX Premixed) (TSQ0101, Qingke, Shanghai, China) was used. Expression levels were normalized to *Actin12* as an internal control. All measurements for each sample were repeated three times. PCR was performed on the CFX384 Touch Real-Time PCR Detection System (CFX384 Touch, Bio-Rad, Hercules, CA, USA). The qRT-PCR program was based on the ArtiCanTM SYBR qPCR Mix manual. The results were exported using CFX Maestro software (version 2.3). The primers used for qRT-PCR are described by Chateigner-Boutin et al. [55]. All primers are listed in Appendix A.

### 4.7. Protein Isolation and Immunoblot Analysis

Total protein extraction was performed as previously described [56]. Protein concentration was determined using the Lowry protein assay kit (F9252, Sigma, Setagaya City, Japan). For immunoblot analysis, total protein was separated by SDS-PAGE and transferred to nitrocellulose membranes (10600001, Cytiva, Marlborough, MA, USA). The membranes were incubated with specific primary antibodies, and signals were detected using the Pro-Light HRP Chemiluminescent Kit (PA112, Tiangen Biotech, Beijing, China). Antibodies against PsaA/B, D1, LHCII, Cytf, PsaN, and LhcaI were generated in-house by expressing and purifying the respective proteins, followed by immunization in rabbits [57].

### 4.8. Transmission Electron Microscopy (TEM) Analysis

To evaluate changes in chloroplast structure, TEM analysis was performed on cotyledons and true leaves from 10-day-old wild-type and *RFCD1*-RNAi-1 seedlings. Leaf tissues were cut into small pieces and fixed in 3% glutaraldehyde in phosphate buffer for 4 h at 4 °C. After washing 3–4 times with phosphate buffer, the samples were post-fixed in 1% osmium tetroxide (OsO_4_) at 4 °C overnight. The samples were dehydrated with ethanol and embedded in epoxy resin at varying concentrations. Ultra-thin sections were prepared using a Reichert OM2 ultramicrotome with a diamond knife. The sections were stained with 2% uranyl acetate (pH 5.0) and 10 mM lead citrate (pH 12.0), and observed using a transmission electron microscope (JEM-1230; JEOL, Akishima, Japan).

### 4.9. RNA-Seq and Data Analysis

Seven-day-old whole seedlings were used for sequencing analysis. RNA quality was assessed using a Thermo NanoDrop 2000 (Thermo, Waltham, MA, USA) and nucleic acid electrophoresis. Each sample had three biological replicates. Sequencing and data analysis were performed by Meggie Biotechnology Co., Ltd., Shanghai, China, using the Illumina HiSeq™ platform. Data were analyzed using Graphpad Prism (v8.0.2, Irvine, CA, USA) and Microsoft Excel 2016. Values were expressed as mean ± SE. Two-tailed unpaired Student’s *t*-tests were used to evaluate differences between WT and RNAi line. A *p*-value of 0.05 was considered significant (* *p* ≤ 0.05, ** *p* ≤ 0.01, *** *p* ≤ 0.001). The free online platform, majorbio cloud platform (https://www.majorbio.com/) (accessed on 3 January 2025), was used to analyze the transcriptomic data.

### 4.10. Phylogenetic Analysis

The full-length amino acid sequences of RFCD1 and other orthologs in plant species were downloaded from NCBI database (https://blast.ncbi.nlm.nih.gov) (accessed on 7 January 2025). The phylogenetic tree was constructed using MEGA software (version 7.0) by the maximum likelihood method [58].

## 5. Conclusions

We identified a PPR protein, RFCD1, in *Arabidopsis*. Complete loss of RFCD1 leads to embryo lethality. Knockdowns of RFCD1 resulted in phenotypic changes, including white cotyledons and yellowish true leaves, an abnormal chloroplast structure, a reduction in chlorophyll content, as well as a decrease in the accumulation of photosynthetic proteins. Compared with the wild type, the transcript levels of PEP-dependent genes were decreased in the *RFCD1*-RNAi seedlings. And the plastid rRNA processing was also defective which in turn may lead to the disruption of plastid ribosome biogenesis and upregulated expression of most ribosomal proteins.

## Figures and Tables

**Figure 1 plants-14-00921-f001:**
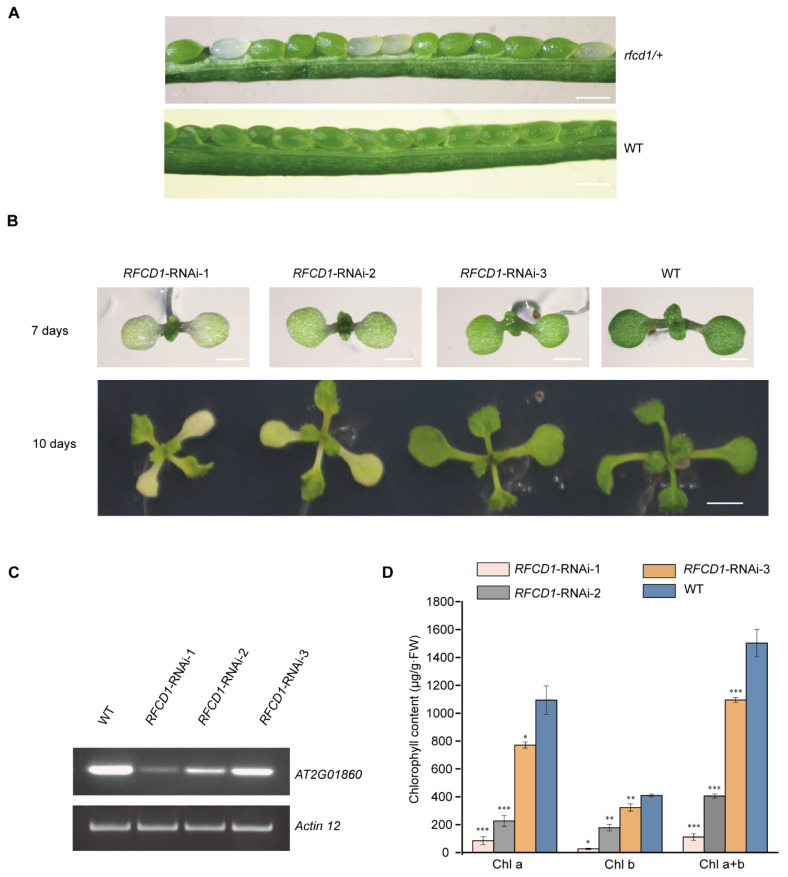
Characterization of the *RFCD1*-RNAi transgenic plants. (**A**) A heterozygous *rfcd1/+* mutant silique showing that approximately one-quarter of the ovules are albino compared to that of wild-type (WT, ecotype Columbia). Scare bars, 0.5 mm. (**B**) Identification and isolation of RNAi lines with different degrees of inhibition of *RFCD1*. Plants were grown on MS medium with 2% (*w*/*v*) sucrose for 7 days and 10 days. Scale bars: 1 mm for 7 days and 2 mm for 10 days. (**C**) Reverse transcription-PCR analysis. RT-PCR was performed using specific primers for AT2G01860 or *ACTIN12* for 35 cycles for WT and RNAi lines with different degrees of decrease in *RFCD1* expression. (**D**) Chlorophyll content of WT and *RFCD1*-RNAi seedlings. Chlorophyll was extracted and quantified from cotyledons of 7-day-old seedlings. Values (means ± SE; n = 3 independent biological replicates) are μg per g fresh weight. The asterisks indicate significant differences between WT and *RFCD1*-RNAi-1. *** *p* < 0.001, ** *p* < 0.01, * *p* < 0.05, by Student’s *t*-test.

**Figure 2 plants-14-00921-f002:**
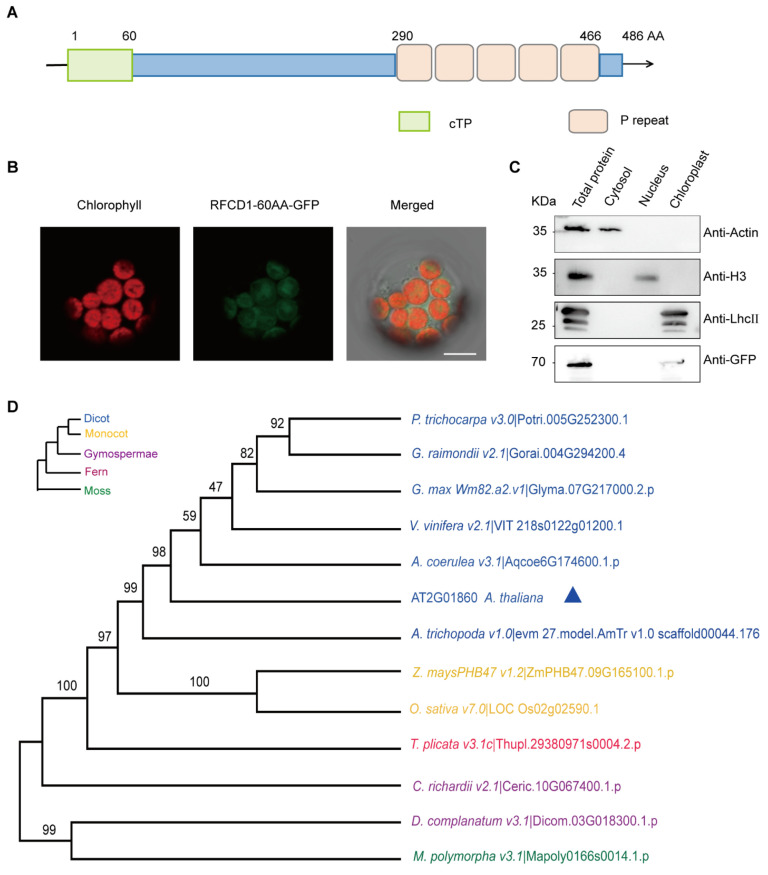
Sequence analysis and subcellular localization of RFCD1. (**A**) Schematic diagram of RFCD1 protein with 5 PPR domains (P). (**B**) Subcellular localization of RFCD1-60AA-GFP. The fluorescence of the RFCD1-60AA-GFP fusion protein in protoplasts was observed using confocal laser scanning microscopy. Green fluorescence signals, chlorophyll red autofluorescence signals, and merged images are shown. Scale bar, 10 μm. (**C**) Subcellular fractionation and immuno-blotting showing the chloroplast localization of RFCD1. Protein extracts of whole cells, cytosol, nucleus, and chloroplast fractions isolated from RFCD1-GFP-transformed *Arabidopsis* protoplasts were subjected to SDS-PAGE, and immuno-blotted using antibodies against GFP (to detect the RFCD1-GFP fusion protein), actin (as a control for cytosol proteins), histone H3 (as a control for nuclear proteins), or LHCII (as a control for chloroplast proteins). (**D**) Phylogenetic tree of RFCD1. All the analyses were performed in MEGA software (version 7.0). *P. trichocarpa*, *Populus trichocarpa*. *G. raimondii*, *Gossypium raimondii*. *G. max*, *Glycine max*. *V. vinifera*, *Vitis vinifera*. *Z. mays*, *Zea mays*. *O. sativa*, *Oryza sativa*. *A. coerulea*, *Arundina coerulea*. *A. thaliana*, *Arabidopsis thaliana. A. trichopoda*, *Arundina trichopoda*. *T. plicata*, *Thuja plicata*. *C. richardii*, *Cyathea richardii*. *D. complanatum*, *Davallia complanatum*. *M. polymorpa*, *Marchantia polymorpha*.

**Figure 3 plants-14-00921-f003:**
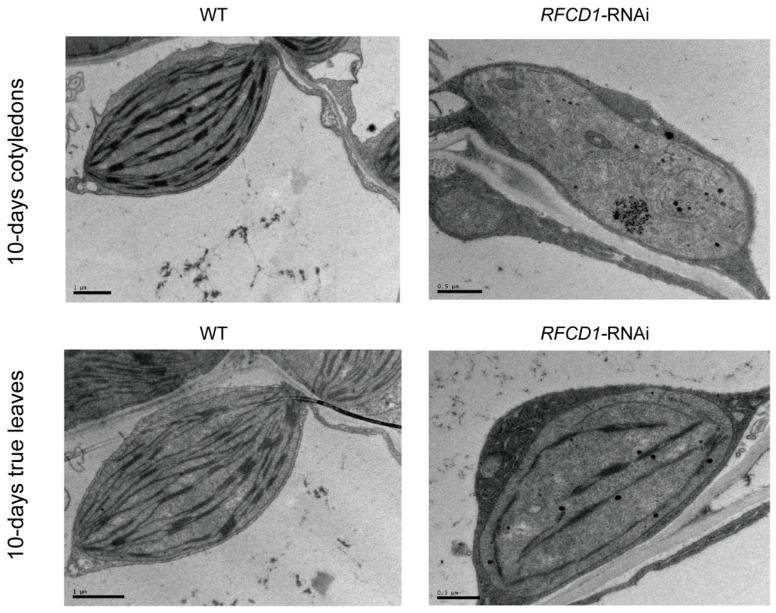
Chloroplast ultrastructure in cotyledons and true leaves from 10-day-old seedlings of the WT and *RFCD1*-RNAi-1 line. Scale bar: 1 µm for WT and 0.5 µm for *RFCD1*-RNAi-1.

**Figure 4 plants-14-00921-f004:**
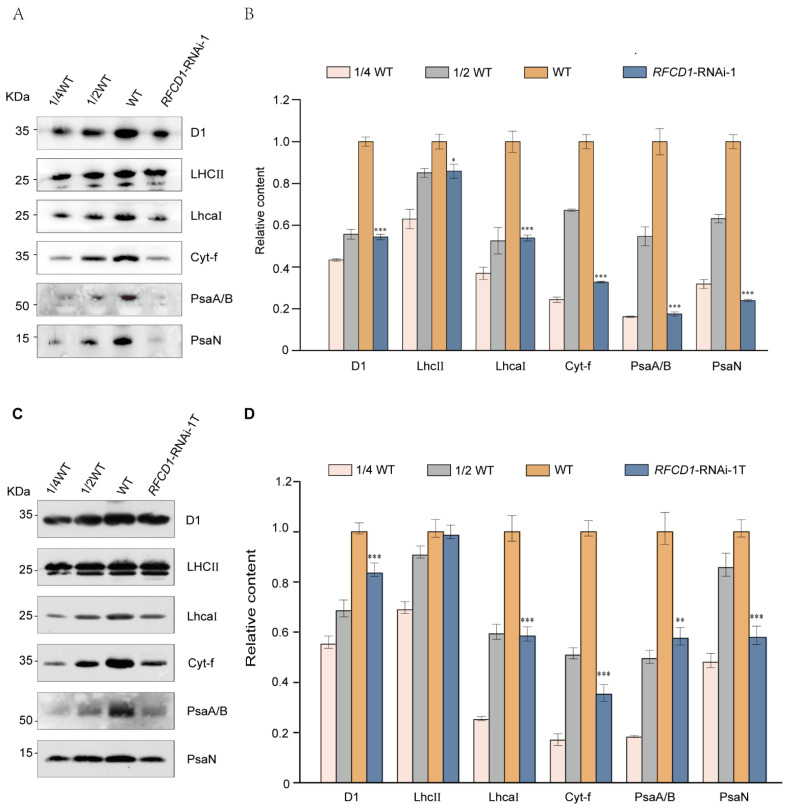
Loss of RFCD1 affects the accumulation of photosynthetic proteins. (**A**) Immunoblot analysis of photosynthetic proteins. Total protein from cotyledons of 7-day-old WT and *RFCD1*-RNAi-1 mutant was separated by 10% Tricine/Sodium dodecyl sulfate polyacrylamide gel electrophoresis (SDS-PAGE) and detected using specific anti-D1, anti-LHCII, anti-LhcaⅠ, anti-Cytf, anti-PsaA/B, and anti-PsaN antibodies. The experiments were repeated at least three times with similar results. (**B**) Proteins immunodetected from (**A**) were analyzed by ImageJ software (version 1.54). Values (means ± SE; n = 3 independent biological replicates) are given as ratios to protein amounts of the WT and *RFCD1*-RNAi-1. (**C**) Immunoblot analysis of photosynthetic proteins. Total protein from whole seedlings of 7-day-old WT and *RFCD1*-RNAi-1 mutant was separated by 10% Tricine/Sodium dodecyl sulfate polyacrylamide gel electrophoresis (SDS-PAGE) and detected using specific anti-D1, anti-LHCII, anti-LhcaⅠ, anti-Cytf, anti-PsaA/B, and anti-PsaN antibodies. The experiments were repeated at least three times with similar results. *RFCD1*-RNAi-1T, samples containing true leaves of *RFCD1*-RNAi-1 mutant. (**D**) Proteins immunodetected from (**C**) were analyzed by ImageJ software (version 1.54). Values (means ± SE; n = 3 independent biological replicates) are given as ratios to protein amounts of the WT and *RFCD1*-RNAi-1. *** *p* < 0.001, ** *p* < 0.01, * *p* < 0.05, by Student’s *t*-test.

**Figure 5 plants-14-00921-f005:**
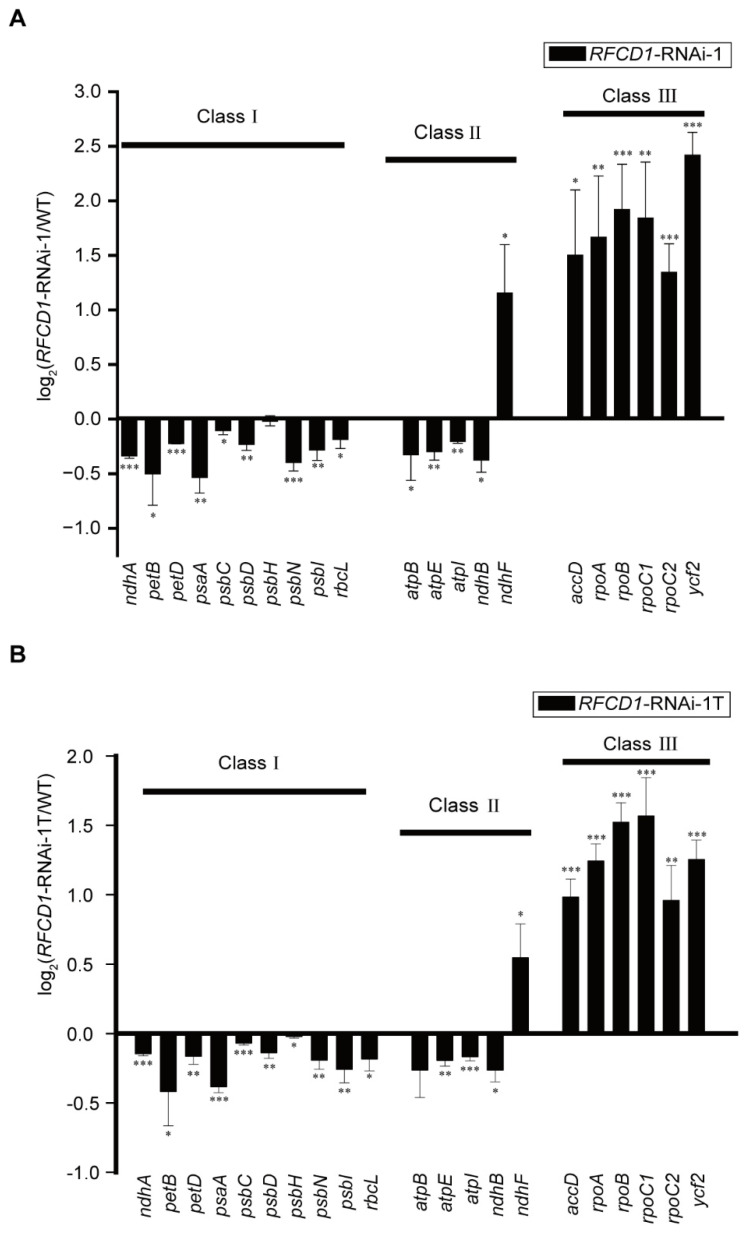
(**A**) Chloroplast gene expression in cotyledons of 7-day-old of *RFCD1*-RNAi-1 compared to that of the WT. Transcript levels of chloroplast genes (Classes I–III) were measured by quantitative RT-PCR. Data (means ± SE; n = 3 independent biological replicates) are given as log_2_ of *RFCD1*-RNAi-1/WT ratios. (**B**) Chloroplast gene expression in true leaves of 10-day-old *RFCD1*-RNAi-1 compared to that of the WT. Transcript levels of chloroplast genes (Classes I–III) were measured by quantitative RT-PCR. Data (means ± SE; n = 3 independent biological replicates) are given as log_2_ of *RFCD1*-RNAi-1T/WT ratios. *** *p* < 0.001, ** *p* < 0.01, * *p* < 0.05, by Student’s *t*-test.

**Figure 6 plants-14-00921-f006:**
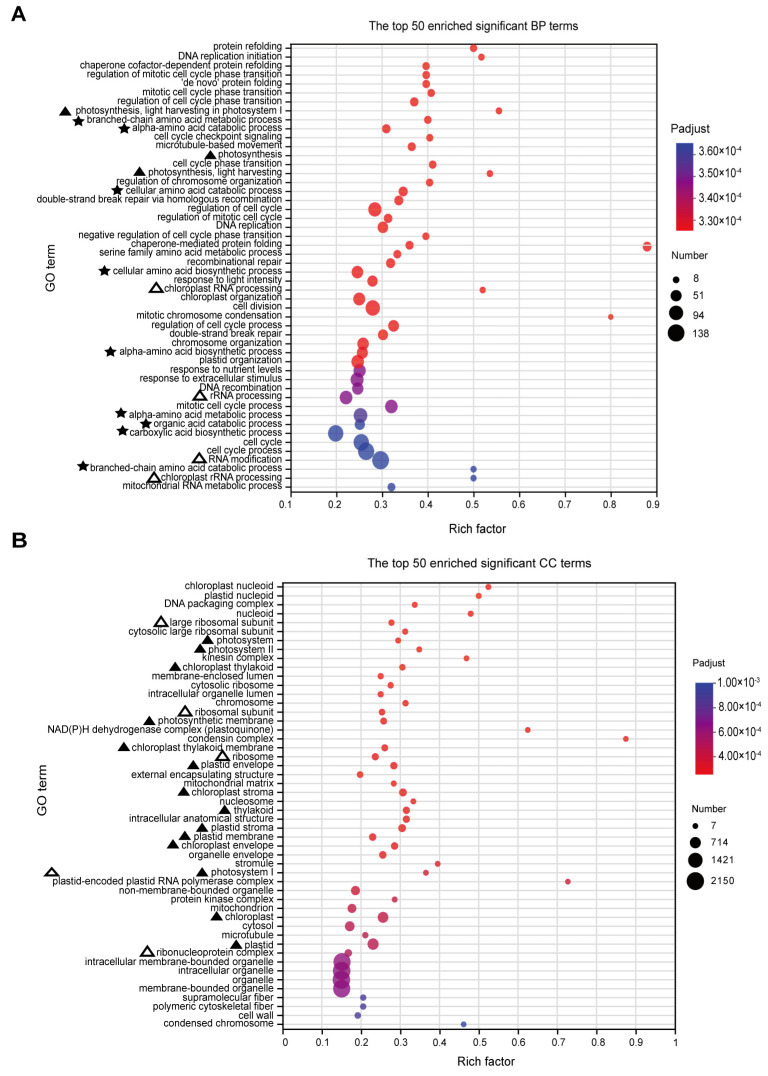
GO enrichment analyses for DEGs. (**A**) The top 50 enriched significant GO BP terms. The terms related to photosynthesis are marked by ‘▲’. The terms related to amino acid metabolism are marked by ‘★’. The terms related to ribosome are marked by ‘△’. (**B**) The top 50 enriched significant GO CC terms. Those related to photosynthesis are marked by ‘▲’. Those related to ribosome are marked by ‘△’. The size of the dots in (**A**,**B**) represents the number of genes included in the GO term. The rich factor in (**A**,**B**) represents the ratio of the number of target genes to the total genes annotated in a pathway. The details of each gene are presented in Appendix A.

**Figure 7 plants-14-00921-f007:**
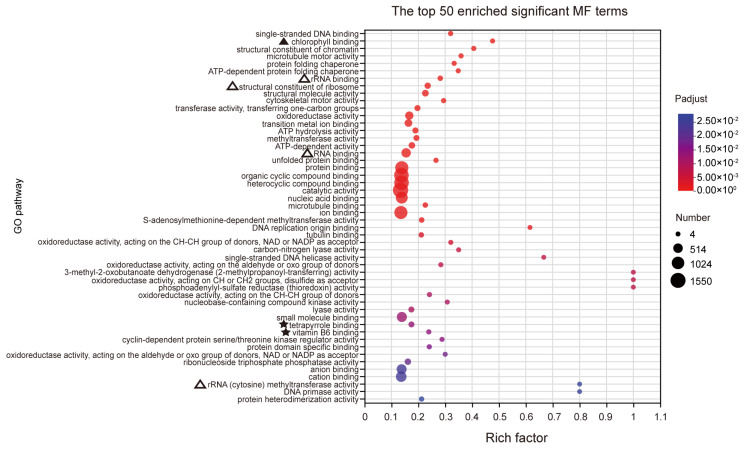
GO analyses for DEGs. The top 50 enriched significant GO MF terms. The terms related to chlorophyll binding are marked by ‘▲’. The terms related to chlorophyll synthesis and vitamin binding are marked by ‘★’. The terms related to ribosome are marked by ‘△’. The details of each gene are presented in Appendix A.

**Figure 8 plants-14-00921-f008:**
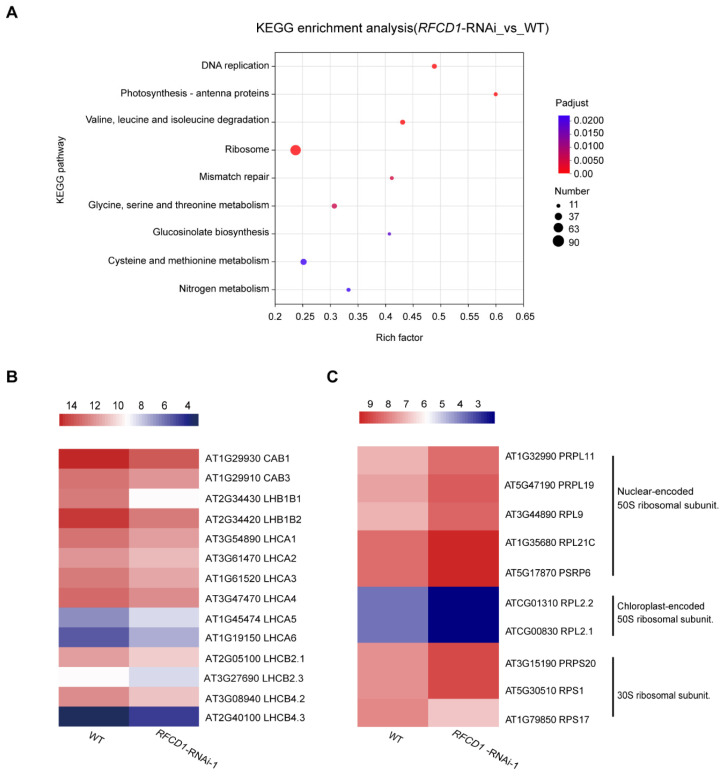
KEGG analyses for DEGs, and heatmap of the DEGs related to photosynthesis–antenna proteins and ribosome. (**A**) The significantly enriched pathways in KEGG pathway analysis. The size of the dots represents the number of genes included in the KEGG term. The rich factor represents the ratio of the number of target genes to the total genes annotated in a pathway. The details of each gene are presented in Appendix A. (**B**) Heatmap of the DEGs enriched in photosynthesis–antenna proteins. (**C**) Heatmap of the DEGs enriched in ribosomal proteins. The bar represents the scale of the expression levels for each gene (log_2_ FPKM) in WT and *RFCD1*-RNAi-1 lines as indicated by rectangles (*p*-adjust ≤ 0.05). The details of each gene are presented in Appendix A.

**Figure 9 plants-14-00921-f009:**
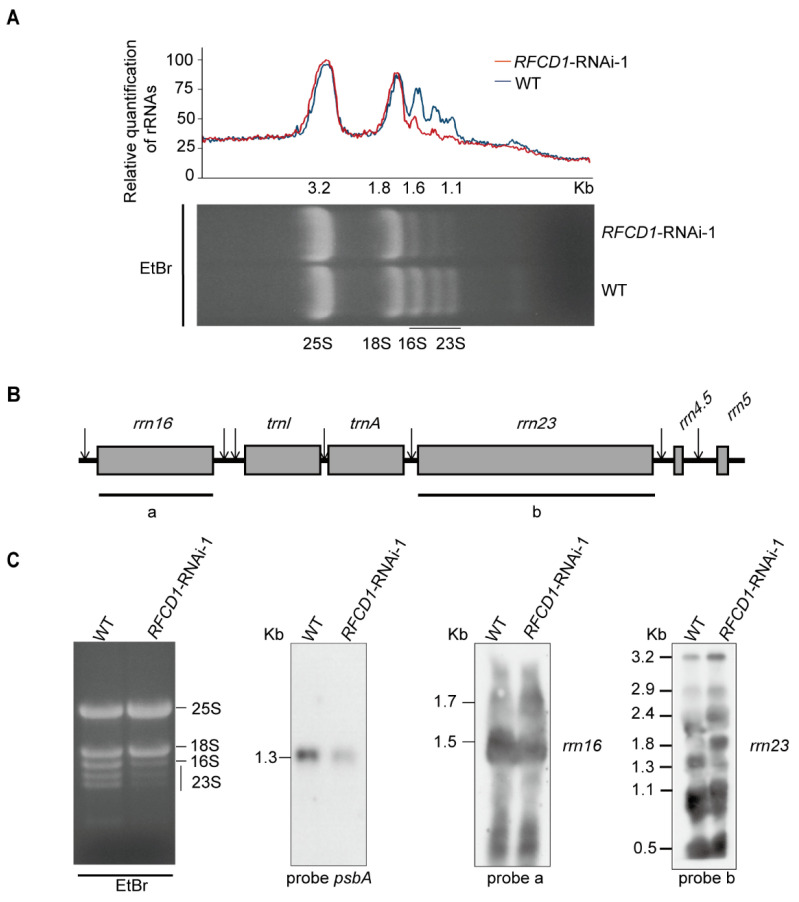
(**A**) Five µg of total RNA from 7-day-old WT and *RFCD1*-RNAi-1 seedlings was separated on a denaturing gel. rRNA content was quantified using ImageJ software (version 1.54). The value of cytoplasmic 25S rRNA in *RFCD1*-RNAi-1 was set to 100, and the relative values of other RNAs were obtained by comparing with 25S rRNA. (**B**) Scheme of the chloroplast *rrn* operon with the two probes used for the RNA blots. The probes for *rrn16*, *rrn23* are marked by black lines under the chloroplast *rrn* operon. a represents the 16S rRNA probe, and b represents the 23S rRNA probe. (**C**) Accumulation of rRNA in WT and *RFCD1*-RNAi-1 seedlings. Total RNA from leaves of 7-day-old WT and *RFCD1*-RNAi-1 seedlings was subjected to RNA blot analysis with specific probes against 16S and 23S rRNAs. Sizes of the distinct forms of the rRNA species are indicated on the left; rRNAs of WT and *RFCD1*-RNAi-1 stained with ethidium bromide were used as loading control.

## Data Availability

Data are available upon reasonable request to the corresponding author.

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
