# Peer review of "A PPR Protein RFCD1 Affects Chloroplast Gene Expression and Chloroplast Development in Arabidopsis"

_plants, 2025, doi:10.3390/plants14060921_

Round 1

Reviewer 1 Report

Comments and Suggestions for Authors

Review of manuscript PLANTS-3467128

The manuscript deals with an interesting topic, the early steps in chloroplast biogenesis during seedling development using Arabidopsis as a model. The authors describe a novel PPR protein (dubbed RFCD1) that becomes seedling lethal if genetically inactivated. By constructing a number of RNAi lines with different degrees of RFCD1 expression they generate plant lines with varying degrees of leaf chlorosis indicating that the protein is indeed required for chloroplast biogenesis. Since the phenotype is largely restricted to cotyledons these mutants belong to the class of snowy cotyledons mutants suggesting that the factor has likely a stage-specific function during seedling development and is not a general chloroplast biogenesis factor. The authors provide a number of analyses of the RNAi lines in order to characterize the impact of the down-regulated protein including electron microscopy of chloroplasts, localisation experiments, pigment and protein determinations as well as western-immuno-blots and RNA-seq data.

The manuscript is largely well written and the data are well presented, however, a number of points require improvements.

The title is an overstatement: The authors do not investigate ribosome assembly or accumulation but only show that plastid rRNA species are highly reduced. The same is true for a number of photosynthesis and other genes. These effects might be caused by indirect influences and the authors cannot conclude that the factor affects ribosome assembly. In fact the true molecular function of the protein remains unknown. Please improve.

The introduction is quite short and somewhat superficial missing to mention essential details such as factors of chloroplast biogenesis like PEP-associated proteins, factors RCB and NCP or  examples for critical PPR proteins, for instance YS1 (all are known to influence CP biogenesis which is highly relevant for the study). Please include references. Further, please include several well accepted overview articles on chloroplast gene expression that provide in depth details relevant for the presented study. In addition, PPR proteins are numerous in plants. Organellar PPRs are mostly involved in editing rather than splicing. Please improve lines 57 and 64.

Results

Lines 76-80: It is not obvious to me how the authors identified the gene for RFCD1. Did they generate CRISPR-cas lines for all chloroplast localized PPR proteins? What are the screening details and the overall strategy? Did other candidates pop-up in this screen and, if yes, why did the authors concentrate on RFCD1?

Lines 113-115: The experiment demonstrates that the predicted transit peptide of RFCD1 is able to navigate GFP into the chloroplast. It however does not provide direct proof of chloroplast localization of RFCD1. Please rephrase or provide additional experimental evidence.

Lines 142-144: TEMs were not done on cotyledons (as for Chl determinations) but with first true leaves. These are less affected (Fig. 1) and are not representative for the effects of RCDF1 in cotyledons. Please provide TEMs for cotyledons to be consistent with other parts of the study.

Lines 157-158: Western analyses were done with 7 day-old plants. Is this done only with cotyledons or with complete seedlings? In the latter case please repeat with cotyledons only as the effects in them most likely are much more pronounced.

Lines 171-172: The comment on PEP and NEP is a repetition of the introduction and can be removed. Again, it is not yet clear whether or not only cotyledons were used for the RNA preparations. Please clarify. If a mixture of cotyledons and true leaves was used, then please repeat with samples isolated from separated tissues.

Line 180: To my knowledge all chloroplast genes could be assigned to a class, please rephrase or clarify what is meant.

The transcriptome analysis was done with 7 day-old seedlings. Again  it is not given which leaves were used for RNA sample preparation. Further, the RNA-seq analysis was done only in duplicate. Triplicate is a minimum for statistical reasons. This part requires thorough additional experimentation and more profound bioinformatical analysis. GO analysis is only a first hint, but all this can be an indirect effect and does not provide conclusive data for the interpretation of the role of RFCD1 in chloroplast biogenesis.

Fig. 9: The reduction in plastid rRNA species (again the tissues source for the RNA preparation is not indicated, please clarify) is very typical for chlorotic leaves and likely not a specific effect of the RFCD1 defect. It is rather a sign for the interrupted chloroplast biogenesis (as reported also for many other albino or chlorotic mutants). Authors should refer to corresponding reviews and databases to obtain a deeper insight into this topic. Conclusions from this section as done in the discussion paragraph (lines436-465) should be taken with more care since a causal relationship is not clear at the prent state.

Author Response

Comments 1: The title is an overstatement: The authors do not investigate ribosome assembly or accumulation but only show that plastid rRNA species are highly reduced. The same is true for a number of photosynthesis and other genes. These effects might be caused by indirect influences and the authors cannot conclude that the factor affects ribosome assembly. In fact the true molecular function of the protein remains unknown. Please improve.

Response 1: Thank you for pointing this out. We agree with this comment. Thus, we have modified the title to “A PPR protein RFCD1 affects chloroplast gene expression and chloroplast development in Arabidopsis”. Please see on the first page of the revised manuscript.

Comments 2: The introduction is quite short and somewhat superficial missing to mention essential details such as factors of chloroplast biogenesis like PEP-associated proteins, factors RCB and NCP or examples for critical PPR proteins, for instance YS1 (all are known to influence CP biogenesis which is highly relevant for the study). Please include references.

Response 2: Thank you for pointing this out. We agree with this comment. Accordingly, we have refined the introduction to provide deeper insights, incorporating key factors involved in chloroplast biogenesis, such as PAPs, RCB and NCP, along with essential PPR proteins like YS1. Please see the introduction section of the revised manuscript that highlighted in red.

Comments 3: Further, please include several well accepted overview articles on chloroplast gene expression that provide in depth details relevant for the presented study.

Response 3: Thank you for pointing this out. We have revised the introduction section based on your suggestions. Please see the introduction section of the revised manuscript that highlighted in red.

Comments 4: In addition, PPR proteins are numerous in plants. Organellar PPRs are mostly involved in editing rather than splicing. Please improve lines 57 and 64.

Response 4: We have revised the subsequent section by modifying the reference to splicing, replacing it with RNA editing in the context of PPR proteins. Please see the introduction section of the revised manuscript that highlighted in red.

Comments 5: Lines 76-80: It is not obvious to me how the authors identified the gene for RFCD1. Did they generate CRISPR-cas lines for all chloroplast localized PPR proteins? What are the screening details and the overall strategy? Did other candidates pop-up in this screen and, if yes, why did the authors concentrate on RFCD1?

Response 5: During the selection of RFCD1, we paid particular attention to determining whether this gene presents a homozygous lethal phenotype. Please see line 126 in the revised manuscript.

Comments 6: Lines 113-115: The experiment demonstrates that the predicted transit peptide of RFCD1 is able to navigate GFP into the chloroplast. It however does not provide direct proof of chloroplast localization of RFCD1. Please rephrase or provide additional experimental evidence.

Response 6: Thank you for pointing this out. We provided an additional fractionation experiment to demonstrate that the RFCD1 protein is localized in the chloroplast. Please see lines 160-165 and Figure 2C.

Comments 7: Lines 142-144: TEMs were not done on cotyledons (as for Chl determinations) but with first true leaves. These are less affected (Fig. 1) and are not representative for the effects of RCDF1 in cotyledons. Please provide TEMs for cotyledons to be consistent with other parts of the study.

Response 7: We provided TEM images of chloroplasts from cotyledons for comparison. Please see Figure 3.

Comments 8: Lines 157-158: Western analyses were done with 7 day-old plants. Is this done only with cotyledons or with complete seedlings? In the latter case please repeat with cotyledons only as the effects in them most likely are much more pronounced.

Response 8: Thank you for pointing this out. Western analyses in lines 157-158 were done with the whole seedlings of 7 day-old plants. So, we performed new western blot analysis for the cotyledons. Please see Figure 4A and 4B.

Comments 9: Lines 171-172: The comment on PEP and NEP is a repetition of the introduction and can be removed. Again, it is not yet clear whether or not only cotyledons were used for the RNA preparations. Please clarify. If a mixture of cotyledons and true leaves was used, then please repeat with samples isolated from separated tissues.

Response 9: Thank you for pointing this out. We agree with this comment. We removed the repetition of description on PEP and NEP in lines 171-172. And cotyledons were used for the RNA preparations. So, we provided new RT-PCR analysis using the true leaves. Please see Figure 5B.

Comments 10: Line 180: To my knowledge all chloroplast genes could be assigned to a class, please rephrase or clarify what is meant.

Response 10: Thank you for pointing this out. We classified them into three categories based on the following criteria according to the reference. The first category consists mainly of genes transcribed by PEP, the second category includes genes transcribed by both PEP and NEP, and the third category consists mainly of genes transcribed by NEP. Please see reference “ECD1 functions as an RNA-editing trans-factor of rps14-149 in plastids and is required for early chloroplast development in seedlings”.

Comments 11: The transcriptome analysis was done with 7 day old seedlings. Again it is not given which leaves were used for RNA sample preparation. Further, the RNA-seq analysis was done only in duplicate.

Response 11: Thank you for pointing this out. We have clearly stated in the manuscript that we used 7-day-old seedlings, and the RNA-seq analysis was performed with three biological replicates. Please see lines 258-260, 264-265 and supplementary figure S4.

Comments 12: Triplicate is a minimum for statistical reasons. This part requires thorough additional experimentation and more profound bioinformatical analysis. GO analysis is only a first hint, but all this can be an indirect effect and does not provide conclusive data for the interpretation of the role of RFCD1 in chloroplast biogenesis.

Response 12: Thank you for pointing this out. We agree with this comment. And we revised the description “Taken together, these results hint that RFCD1 may play a role in the expression of RNA processing and ribosomal genes, and genes related to the fundamental metabolic processes specially or mainly taken place in chloroplast, such as photosynthesis and amino acid metabolism.”. Please see lines 306-309.

Comments 13: Fig. 9: The reduction in plastid rRNA species (again the tissues source for the RNA preparation is not indicated, please clarify) is very typical for chlorotic leaves and likely not a specific effect of the RFCD1 defect. It is rather a sign for the interrupted chloroplast biogenesis (as reported also for many other albino or chlorotic mutants). Authors should refer to corresponding reviews and databases to obtain a deeper insight into this topic. Conclusions from this section as done in the discussion paragraph (lines436-465) should be taken with more care since a causal relationship is not clear at the prent state.

Response 13: Thank you for pointing this out. We have referred to corresponding reviews to obtain a deeper insight into this topic. Please see the introduction section of the revised manuscript that highlighted in red. And we also added several examples that loss-of-function in some PPR or OPR proteins lead to the interrupted chloroplast biogenesis (as reported also for albino or chlorotic phenotype). Please see the discussion section of the revised manuscript that highlighted in red.

Reviewer 2 Report

Comments and Suggestions for Authors

The manuscript titled "RFCD1 Affects Early Chloroplast Development and Ribosome Assembly in Arabidopsis" presents a comprehensive and well-structured study on the role of the novel PPR protein, RFCD1, in chloroplast development. The authors have employed a combination of fluorescence localization analysis, knockout mutant characterization, RNA interference, qRT-PCR, northern blotting, and RNA sequencing to elucidate the function of RFCD1. The insights expanded from this research significantly advance our understanding of the molecular mechanisms governing chloroplast development. I recommend the acceptance of this manuscript for publication.

Author Response

Thank you for your comments.

Round 2

Reviewer 1 Report

Comments and Suggestions for Authors

Review of revised manuscript PLANTS-3467128

The authors improved the manuscript largely according to my suggestions, but there are some issues left.

The introduction has been clearly improved, for a more balanced citation style please include all four cryoEM studies on the PEP complex that occurred in 2024. The description of the modules differs between these four, so pay attention to adapt your description.

I found only one new review citation on chloroplast gene expression, but none from the leading labs in the field. Again, for reasons of balance include further reviews preferentially those that discuss chloroplast development and chlorotic phenotypes which puts your own study in a broader context.

The explanation how the authors identified RFCD1 is very poor and the revision in line 126 includes only the lethality comment. Please explain in the text your screening strategy in a way that the interested reader can follow it.

The RNAseq analysis should be done in triplicate, but the method description in line 575 says that each sample had only duplicate repetitions, please clarify.

Comments on the Quality of English Language

In general, the English is fine, there are however at some places in the manuscript problems with the determined and undetermined articles I recommend to ask an native speaker to improve this.

Author Response

Comments 1: The introduction has been clearly improved, for a more balanced citation style please include all four cryoEM studies on the PEP complex that occurred in 2024. The description of the modules differs between these four, so pay attention to adapt your description.

Response 1: Thank you for pointing this out. We agree with this comment. We have re-written this part according to the Reviewer’s suggestion. Please see lines 49-67.

Comments 2: I found only one new review citation on chloroplast gene expression, but none from the leading labs in the field. Again, for reasons of balance include further reviews preferentially those that discuss chloroplast development and chlorotic phenotypes which puts your own study in a broader context.

Response 2: Thank you for pointing this out. We agree with this comment. As suggested by the reviewer, we have added new references to improve the introduction. Please see lines 88-90, 101-103, 109-114, 118-120.

Comments 3: The explanation how the authors identified RFCD1 is very poor and the revision in line 126 includes only the lethality comment. Please explain in the text your screening strategy in a way that the interested reader can follow it.

Response 3: Thank you for pointing this out. This study focuses on members of the chloroplast PPR protein family, whose loss of function leads to an embryo-lethal phenotype, compromising their roles in chloroplast development. To address this, we first validated all unknown chloroplast PPR proteins via the CRISPR-Cas9 system to confirm the reproducibility of the embryo-lethal phenotypes in the obtained knockout mutants. Based on this, we selected some genes to construct knockdown mutants via RNA interference, thereby circumventing the embryonic lethality caused by the complete knockout. Finally, we investigated the RNAi mutants defective in early chloroplast development for further study.

Comments 4: The RNAseq analysis should be done in triplicate, but the method description in line 575 says that each sample had only duplicate repetitions, please clarify.

Response 4: Thank you for pointing this out. The statement " each sample had only duplicate repetitions" in the original text was an incorrect. In fact, all critical experiments strictly followed a standard protocol of three independent biological replicates to ensure data reliability. We have re-examined the raw data, and all conclusions are supported by results from three replicates, with statistical analyses (e.g., standard deviation, p-values) confirming the robustness of the findings. The description has been corrected to "three independent replicates" in Line 598 of the revised manuscript. We sincerely apologize for this oversight and have thoroughly reviewed similar statements in the revised manuscript to prevent such errors. This correction does not affect the core conclusions of the study. Thank you for helping us improve the quality of the paper.

Comments 5: In general, the English is fine, there are however at some places in the manuscript problems with the determined and undetermined articles I recommend to ask an native speaker to improve this.

Response 5: Thanks for your suggestion. We have invited Professor Jean-David Rochaix to polish the language. We have highlighted the words and sentences in red.